# Potential Therapeutic Skin Microbiomes Suppressing *Staphylococcus aureus*-Derived Immune Responses and Upregulating Skin Barrier Function-Related Genes via the AhR Signaling Pathway

**DOI:** 10.3390/ijms23179551

**Published:** 2022-08-23

**Authors:** Eulgi Lee, Kyungchan Min, Hyeok Ahn, Bu-nam Jeon, Shinyoung Park, Changhee Yun, Hyehee Jeon, Jae-sung Yeon, Hyun Kim, Hansoo Park

**Affiliations:** 1Department of Biomedical Science and Engineering, Gwangju Institute of Science and Technology (GIST), Gwangju 61005, Korea; 2Genome and Company, Pangyo-ro 255, Bundang-gu, Seoungnam-si 13486, Korea

**Keywords:** *Staphylococcus aureus*, aryl hydrocarbon receptor, skin barrier

## Abstract

Disruption of the skin microbial balance can exacerbate certain skin diseases and affect prognosis and treatment. Changes in the distribution and prevalence of certain microbial species on the skin, such as *Staphylococcus aureus* (SA), can impact the development of severe atopic dermatitis (AD) or psoriasis (Pso). A dysfunctional skin barrier develops in AD and Pso due to SA colonization, resulting in keratinization and chronic or progressive chronic inflammation. Disruption of the skin barrier following SA colonization can elevate the production of T helper 2 (Th2)-derived cytokines, which can cause an imbalance in Th1, Th2, and Th17 cells. This study examined the ability of potential therapeutic skin microbiomes, such as *Cutibacterium avidum* R-CH3 and *Staphylococcus hominis* R9, to inhibit SA biofilm formation and restore skin barrier function-related genes through the activation of the aryl hydrocarbon receptor (AhR) and the nuclear factor erythroid-2-related factor 2 (Nrf2) downstream target. We observed that IL-4/IL-13-induced downregulation of *FLG*, *LOR*, and *IVL* induced by SA colonization could be reversed by dual AhR/Nrf2 activation. Further, *OVOL1* expression may be modulated by functional microbiomes via dual AhR/Nrf2 activation. Our results suggest that our potential therapeutic skin microbiomes can prevent SA-derived Th2-biased skin barrier disruption via IL-13 and IL-4-dependent *FLG* deregulation, STAT3 activation, and AhR-mediated STAT6 expression.

## 1. Introduction

The bacterial community on the skin surface is essential for maintaining a functional microbiome. The disruption of the skin microbial balance (dysbiosis) in certain skin diseases can not only significantly affect the severity of the disease but can also be important for prognosis and treatment [1,2]. Specifically, it has been discovered that changes in the distribution of some skin microbial species and the predominance of certain species can significantly influence the progression of some skin diseases [3]. For example, *Staphylococcus aureus* (SA) is found on the skin of patients with atopic dermatitis (AD) at the site of lesions caused by AD, and many studies report that it acts as a super-antigen against T cells, impeding the function of the skin barrier or activating the inflammatory response [4,5]. Furthermore, psoriasis (Pso) is exacerbated by immune dysregulation and skin colonization by SA, which alters the inflammatory response and contributes significantly to the severity of Pso [6,7]. In this way, SA colonization leads to the increasing predominance of the species, which is common to both diseases [8]. As a result of SA colonization, the biofilm produced in the skin can damage the epidermal barrier, cause congenital and acquired immune regulation disorders, and cause chronic inflammatory cytokines in many skin diseases, including AD, Pso, and acne [9,10,11].

SA colonization leads to skin lesions with a dysfunctional skin barrier due to the reduction of β-defensin and cathelicidin [12,13]. Particularly, filaggrin (*FLG*) deficiency induced by the formation of SA biofilms and suppression of involucrin (*IVL*) and loricrin (*LOR*) gene expression in the skin barrier can lead to chronic exposure to SA and accelerate chronic AD or Pso [9]. This can also lead to disruption of the skin barrier, triggering abnormal immune responses through T-helper 2 cells (Th2), the release of pro-inflammatory cytokines from keratinocytes, and the stimulation of mast cell degranulation [14,15]. Skin inflammation caused by skin barrier disruption is accompanied by increased levels of Th2 cytokines, such as interleukin (IL)-4 and IL-13, which suppress the expression of antimicrobial peptides [16,17]. IL-4/IL-13-induced responses can be driven by signal transducers and transcriptional activators (STAT) 6 and promote thymic stromal lymphopoietin (TSLP) production, a cardinal keratinocyte-derived cytokine that contributes to Th2 immunity [18,19]. Subsequently, epidermal keratinocytes of the skin, whose barrier is damaged, create large amounts of TSLP, IL-25, and IL-33, which stimulate the production of IL-13, forming a vicious cycle of conditions that promote AD [18,20,21]. Furthermore, chronic skin colonization with SA can result in increased levels of IL-17, triggering the release of pro-inflammatory cytokines, such as IL-6 and IL-8 [22,23]. During chronic or late stages of AD, Th1, Th2, and Th17 cell ratios are out of balance and cause them to interact improperly [23,24]. As a result of Th17 cell infiltration, epithelial cells produce IL-22 and IL-17A, leading to tissue fibrosis, chronic inflammation, and progression of chronic inflammatory skin lesions [24,25,26].

To determine the possibility of skin barrier function recovery, we assessed the effects of skin microbiotas that inhibit SA growth and biofilm development on skin barrier function recovery through the aryl hydrocarbon receptor (AhR)/nuclear factor erythroid-2-related factor 2 (Nrf2) pathway. The dual activity of AhR and Nrf2 has been shown to restore the downregulation of FLG, LOR, and IVL caused by IL-4/IL-13. IL-4 and IL-13 expression and subsequent increases in STAT6 phosphorylation occur in response to oxidative stress [27]. Keratinocytes activate Nrf2 and upregulate NAD(P)H quinone oxidoreductase-1 to neutralize IL-4/IL-13-induced reactive oxygen species (ROS) or heme oxygenase-1 to inhibit STAT6 phosphorylation [28]. Furthermore, oxidative stress decreases the expression of skin barrier factors by activating STAT3 [29]. Thus, it is proposed that Nrf2 signaling can reduce oxidative stress in keratinocytes and inhibit STAT3, thereby preventing skin barrier factor suppression. The Nrf2 signaling pathway reduces immune responses by inhibiting IL-4/IL-13-induced FLG downregulation, IL-13-induced STAT3 activation, and IL-4/STAT6-induced Th2 cell differentiation, IgE production, and chemokine expression [30,31]. Activation of AhR leads to upregulation of OVOL1, a crucial upstream transcription factor involved in FLG and LOR signaling in addition to Nrf2 signaling [32]. As a result, OVOL1 nuclear translocation can potentially help restore the downregulation of skin barrier factors, such as IVL, LOR, and FLG, induced by IL-4 and IL-13 through AhR activation [33,34].

Disruption of the skin barrier associated with SA colonization on the skin can partially explain the clinical and pathophysiological similarities between AD and Pso. To investigate this association, this study aimed to identify microbes that control skin barrier function-related genes to prevent SA colonization. Identifying potential therapeutic microbiomes that inhibit SA colonization will provide novel insights into the common pathological mechanisms underlying SA-associated AD and Pso, as well as guide the development of therapies targeting metabolites identified from these potential therapeutic microbiomes.

## 2. Results

### 2.1. Staphylococcus hominis and Cutibacterium avidum Significantly Inhibit the Growth of SA-Derived Biofilms

To determine the specific microbial features of AD and Pso, we compared the microbial composition of AD and Pso lesion sites with those of healthy controls. We used an open dataset from a study by Tsoi et al. in the Gene Expression Omnibus (GEO) database. The dataset contains 147 human skin whole transcriptome sequences, which includes 38 healthy controls, 27 patients with AD, and 28 patients with Pso [35] (Figure 1a). Lesions in patients with AD were enriched in SA, while *S. hominis* and *Kocuria palustris* were enriched in healthy controls (Appendix A). Even though SA was not significantly enriched in the lesions of Pso patients like in the AD cohort, *Staphylococcus hominis* was considerably more prevalent among healthy controls than in patients with Pso. Furthermore, *Cutibacterium avidum* was enriched in the healthy controls compared to Pso lesion sites (Appendix A).

We isolated *S. hominis*, *K. palustris*, and *C. avidum* from donors without a history of skin disease to examine their ability to inhibit the formation of SA biofilms (Table 1). First, we investigated the ability of *S. hominis* and *K. palustris* to inhibit the formation of SA biofilms from the AD group. Both *S. hominis* and *K. palustris* considerably inhibited the formation of biofilms in SA from the AD group. However, *S. hominis* inhibited SA biofilm formation more than *K. palustris* (Figure 1b). Next, we investigated whether *C. avidum*, detected considerably in healthy controls compared to the Pso cohort, could inhibit SA biofilm formation in addition to the two bacteria found in the AD cohort. There was a considerable inhibition of SA biofilm formation by all four isolates of *C. avidum*, but the *C. avidum* R-CH3 strain showed the most significant inhibition (Figure 1c).

We also compared the Pso group SA biofilm formation and growth-inhibiting abilities of the three species, *S. hominis*, *K. palustris*, and *C. avidum*. SA biofilm formation was inhibited by all three species, but *K. palustris* was significantly less effective in inhibiting SA biofilm formation than the other two species. Further, compared to the baicalein set as a positive control, *K. palustris* did not significantly inhibit the formation of SA biofilms (Figure 1d).

As a final step, we performed the overlay clean zone test to determine whether the three species of bacteria can directly inhibit the growth of SA. *K. palustris* showed significantly lower SA growth inhibition activity compared to the other two bacterial species. Even though the SA growth inhibitory activity of *S. hominis* was slightly higher than that of *C. avidum*, it was practically identical to that of triclosan (positive control) (Figure 1e).

### 2.2. Characteristics of AD- and Pso-Induced Immune Alterations in Response to the Reduction in S. hominis Abundance

We observed that *S. hominis* was significantly more abundant in controls than in AD and Pso lesions, and that *S. hominis* inhibited SA biofilm formation and direct growth of SA. Accordingly, we divided *S. hominis* into high and low abundance groups. Using the tool Kraken2, zero counts were considered low abundance, and positive counts were considered high abundance. We then performed gene set enrichment analysis (GSEA) to determine differentially expressed genes (DEGs), their related signaling pathways, and their biological significance. We observed that in the AD cohort, the interferon-gamma (IFN-γ) response set (normalized enrichment score [NES] = −2.290, nominal [NOM] *p* < 0.001 and FDR *q*-value < 0.001), TNF-α signaling through NF-κb signaling (NES = −2.225, NOM *p* < 0.001 and FDR *q*-value < 0.001), IL6-JAK-STAT3 signaling (NES = −2.182, NOM *p* < 0.001 and FDR *q*-value < 0.001), and inflammatory response (NES = −1.763, NOM *p* < 0.001 and FDR *q*-value = 0.001) were enriched in the *S. hominis* low group (Figure 1f and Appendix A). We also analyzed differences in gene set enrichment under identical conditions to examine DEGs in the Pso cohort. As in the AD group cohort, NES demonstrated that the IFN-γ response set (NES = −2.576, NOM *p* < 0.001 and FDR *q*-value < 0.001), TNF-α signaling through NF-κb signaling (ES = −0.451, NES = −1.854, NOM *p* < 0.001 and FDR *q*-value < 0.001), IL6-JAK-STAT3 signaling (NES = −2.143, NOM *p* < 0.001 and FDR *q*-value < 0.001) and inflammatory response (NES = −1.964, NOM *p* < 0.001 and FDR *q*-value < 0.001) were significantly higher in the *S. hominis* high group (Figure 1f and Appendix A).

To examine the composition of immune cells among the *S. hominis* high versus low groups, digital cytometry was performed using the CIBERSORTx platform [36]. We found that the *S. hominis* high group exhibited higher expression of resting mast cells than the *S. hominis* low group in both AD and Pso. Further, follicular Th cells capable of activating B cells were significantly lower in the *S. hominis* high group. In the *S. hominis* high group, the level of activated dendritic cells (DCs) was characteristically lower in AD than in the *S. hominis* low group. Moreover, the level of neutrophils and macrophages was lower in Pso than in the *S. hominis* high group (Figure 1g). These findings suggest that *S. hominis* can potentially reduce the onset of immune responses and tissue damage by reducing signs and symptoms similar to those caused by allergic reactions to SA bioproducts.

### 2.3. Immune Suppression Mediated by SA-Derived Mechanisms in C. avidum Is Similar to S. hominis

GSEA was performed in the Pso cohort to investigate the DEG-related signaling pathways and to determine if immune responses are similar between *S. hominis* and *C. avidum* enrichment levels. GSEA showed that the *C. avidum* low group had inflammatory responses comparable to the *S. hominis* low group, as well as increased INF-α response, INF-γ response, TNF-α signaling via NF-κb signaling, and IL6-JAK-STAT3 signaling (Figure 2a). Even in the *C. avidum* low group, the IFN-γ response set (NES = −2.619, NOM *p* < 0.001 and FDR *q*-value < 0.001), TNF-α signaling via NF-κb signaling (NES = −2.059, NOM *p* < 0.001 and FDR *q*-value < 0.001), IL6-JAK-STAT3 signaling (NES = −2.235, NOM *p* < 0.001 and FDR *q*-value < 0.001), and inflammatory response (NES = −2.148, NOM *p* < 0.001 and FDR *q*-value < 0.001) were consistent with those seen in the *S. hominis* low group in AD as well as Pso (Figure 2b). Furthermore, the heatmap of the Pso cohort also showed that immune-related genes, including those involved in IL6-JAK-STAT3 signaling, INF-γ signaling, and TNF-α signaling, were expressed at higher levels in the *C. avidum* low group than in the *C. avidum* high group (Figure 2c).

Next, we analyzed the composition of immune cells between the high and low *C. avidum* groups using digital cytometry. In the *C. avidum* high group, resting mast cells were significantly more abundant than in the *C. avidum* low group, and macrophages were significantly less abundant than in the *C. avidum* low group. Furthermore, the relative abundance of DCs was significantly lower in the *C. avidum* low group than in the high group (Figure 2d). Taken together, these results suggest that both *S. hominis* and *C. avidum* are capable of inhibiting the formation of SA biofilms. In addition, similarities were observed in their immune response mechanisms in accordance with their bacterial abundance.

We then analyzed the expression of interleukin (IL)-25, IL-33, and TSLP to confirm that previously screened samples, such as *S. hominis* R9 and *C. avidum* R-CH3, inhibited SA biofilm formation but also inhibited the induction of immune responses. *IL-25*, *IL-33*, and *TSLP* mRNA expression increased from a minimum of 8-fold to a maximum of 13-fold in SA-treated HaCaT cells compared to normal HaCaT cells. Alternatively, when SA-treated HaCaT cells were treated with *S. hominis* R9 and *C. avidum* R-CH3, significant reductions in mRNA levels of *IL-25*, *IL-33*, and *TSLP* were observed in both groups compared to the SA-alone treated group (Figure 2e). Moreover, *IL-5* expression that increased with SA treatment significantly reduced after treatment with *S. hominis* R9 and *C. avidum* R-CH3 supernatant. This reduction in *IL-5* expression was more significant in the *C. avidum* R-CH3 treated group compared to *S. hominis* R9 (Figure 2f). In addition, *IL-31* expression was also elevated during SA treatment, but *S. hominis* R9 and *C. avidum* R-CH3 significantly reduced this expression, and *C. avidum* R-CH3 significantly reduced IL-31 levels relative to *S. hominis* R9 (Figure 2g). Hence, *S. hominis* R9 and *C. avidum* R-CH3 inhibit the induction of Th2 cell-mediated immune responses by interfering with SA colonization, suggesting a possible reduction in B-cell activation and inhibition of allergen-specific Ig E production [37].

### 2.4. The C. avidum R-CH3 Strain Exerts Greater Effects on SA-Derived Th2-Biased Cytokines than the S. hominis R9 Strain

The hyperproliferation of SA results in increased levels of IL-4 and IL-13, which ultimately destroys the skin barrier due to mast cell degeneration and Th2 cell bias [11]. To determine the effects of IL-4 and IL-13, we divided the high and low abundance groups of *S. hominis* and *C. avidum* based on *IL-4R* and *IL-13R1* expression. The function of heterodimers IL-4Ra and IL-13Ra1 in response to IL-4/IL-13 is to activate Janus kinase 2 (JAK2) and tyrosine kinase 2 (TYK2), as well as signal transducer and activator of transcription (STAT)3 and STAT6 [38,39].

We found that in both AD and Pso cohorts, the expression rate of the *IL4R* gene was significantly lower in the high abundance group of *S. hominis* or *C. avidum* compared to the low abundance group. Furthermore, the expression of *IL-13Ra1* was significantly lower in the *C. avidum* high group in the Pso cohort. In the *S. hominis* high group, *IL-13Ra1* expression was lower in AD and Pso cohorts than in the *S. hominis* low group. However, these differences were not statistically significant (Figure 3a).

As a critical cytokine involved in the development of allergic inflammation, IL-4 is primarily responsible for transforming naive T cells into Th2 lymphocytes through the generation of many effector cytokines, including IL-5, IL-6, and IL-13 [40,41]. SA increased *IL-4* expression more than six-fold, and *S. hominis* R9 and *C. avidum* R-CH3 significantly reduced *IL-4* expression following SA treatment. Additionally, the *C. Avidum* R-CH3 treatment group significantly reduced *IL-4* expression to a greater extent than the *S. hominis* R9 treatment group (Figure 3b).

In addition, IL-13 promotes the development of atopic inflammation via IL-4R𝛂/IL-13R𝛂1 and contributes to the activation of STAT6/STAT3 by activating downstream JAK1/TYK2/JAK2 [42]. Together with IL-4, IL-13 can induce Th2-biased T cell differentiation, IgE production in B cells, and Th2-associated chemokines, such as CCL17 and CCL22 in dendritic cells (DCs) [27]. In response to SA, the expression of *IL-13* significantly increased, while the expression of *IL-13* with *S. hominis* R9 and *C. avidum* R-CH3 treatment significantly decreased. On the other hand, it was observed that treatment with *C. avidum* R-CH3 significantly reduced the expression of SA-derived *IL-13* to a greater extent than *S. Hominis* R9 treatment (Figure 3c). Furthermore, both *S. Hominis* R9 and *C. Avidum* R-CH3 significantly reduced the production of SA-derived *IL-6*. Interestingly, we also observed that *C. Avidum* R-CH3 significantly reduced SA-induced expression of *IL-6* to a greater extent compared to *S. hominis* R9 (Figure 3d). Therefore, not only did SA significantly enhance the expression of *IL-24*, and both *S. hominis* R9 and *C. avidum* R-CH3 significantly reduced the expression of SA-derived *IL-24*, but *C. avidum* R-CH3 was once again shown to significantly reduce *IL-24* expression to a greater extent than *S. hominis R9* (Figure 3e). Furthermore, it was shown that *STAT3* and *STAT6* significantly increased in SA-treated AD and Pso groups. It was established that *S. hominis* and *C. avidum* contributed to the effective reduction of STAT3/STAT6. For both STAT3 and STAT6, *C. avidum* R-CH3 showed a significantly greater decrease compared to the SA-only treated group and the *S. hominis* R9 treated group (Figure 3f).

We also examined IL-17/IL-22 signaling and chronic SA colonization to determine whether chronic AD or Pso could progress due to chronic SA colonization. *IL-17A*, *IL-17F,* and *IL-22* all showed significant increases in their mRNA expression during SA treatment. Additionally, compared to the SA-treated group, both *S. hominis* R9 and *C. avidum* R-CH3 treatment groups showed a significant decrease in IL-17A/F and IL-22 production (Figure 3g). After verifying the activation potential of Th1, Th17, and Th22 cells according to an increase in IL-17/IL-22 signals induced by chronic SA colonization, treatment with *S. hominis* R9 and *C. avidum* R-CH3 was examined to determine if the protein levels of IFN-γ, TNF-α, and IL-1β could be significantly decreased upon treatment with these bacterial species. A significant increase in IFN-γ, TNF-α, and IL-1β protein expression was observed following SA treatment, but their expression was significantly decreased after treatment with *S. hominis* R9 and *C. avidum* R-CH3. Additionally, *C. avidum* R-CH3 showed a significantly lower increase in protein expression than the *S. hominis* R9 treated group (Figure 3h).

### 2.5. C. avidum R-CH3 Can Inhibit Intracellular ROS-Induced Apoptosis via AhR/Nrf2 Dual Signaling

We examined whether AhR-mediated signaling can reduce the generation of reactive oxygen species (ROS) by attenuating Th2-biased skin inflammation caused by increased SA colonization. Using the SA abundance of the entire dataset, we divided groups by high versus the low abundance of SA. We then plotted the AhR signal, Nrf2 signal, and gene sets corresponding to skin barrier functions on a heat map (Figure 4a). Neither SA high nor low groups significantly increased *AHR* gene expression, but the SA high group showed a significant increase in AhR-nuclear translocator (*ARNT*) and cytochrome P450 (*CYP1*) family gene expression. In addition, both *NFE2L2*, as well as downstream *HMOX1* and *NQO1*, were significantly increased in the SA low group. In the SA high group, an increased expression of *FLG*, *LOR*, and *CLDN1* was observed. We further investigated the accumulation of ROS in skin keratinocytes following SA colonization by analyzing the expression levels of genes involved in classical AhR signaling, such as *AHR*, *ARNT*, *CYP1A1*, and *PTGS2* (Figure 4b). A slight reduction in *AHR* expression was observed in the *C. avidum* R-CH3 treatment group, but no significant differences were observed between the *S. hominis* R9 treatment and SA-alone groups. The *C. avidum* R-CH3 treatment group also demonstrated a small decrease in *AHR* expression compared to the *S. hominis* R9 treatment group.

The expression of *ARNT* and *CYP1A1*, downstream genes regulated by AhR activity, were significantly increased during SA treatment; however, their expression significantly decreased in the *S. hominis* R9 and *C. avidum* R-CH3 treatment groups. The *C. avidum* R-CH3 treatment group had a significantly greater decrease in *ARNT* and *CYP1A1* expression than the *S. hominis* R9 treatment group, consistent with previous analyses. *PTGS2*, which encodes the cyclooxygenase-2 (Cox-2) protein, is also involved in AhR activation induced by various stimuli, such as growth factors and cytokines [43]. SA significantly increased the expression of *PTGS2*, whereas both *S. hominis* R9 and *C. avidum* R-CH3 treatment groups significantly decreased the expression of *PTGS2* after SA treatment. As with CYP1A1, *C. avidum* R-CH3 significantly reduced the expression of *PTGS2* to a greater extent than the S. hominis R9 treatment group. Taken together, activation of AhR signaling by SA may contribute to the increased expression of CYP1A1 and PTGS2, as well as the generation of ROS.

Accumulation of ROS in cells can lead to apoptosis due to MAPK signaling [44,45]. By correlating the results of the previous experiments with transcriptome analyses (Figure 2b,c), we evaluated the levels of gene expression of *TNFAIP8*, *TNIP*, and *NFκB1* in *S. hominis* and *C. avidum* by abundance grouping (Appendix A). There was a notable decrease in the expression of all three genes when the abundance of *C. avidum* decreased. We next examined the Bax/Bcl-2 ratio in the *S. hominis* R9 and *C. avidum* R-CH3-treated groups compared to the SA-only treated group to determine the degree of apoptosis induced by ROS (Appendix A). Both microbiotas significantly reduced the Bax/Bcl-2 ratio, which could be used as a proxy for reducing SA-induced apoptosis. The expression of *Bcl-2* did not significantly change in the *C. avidum* R-CH3 treated group, and the expression of *Bax* was markedly reduced compared to the *S. hominis* R9 treated group, indicating a significant decrease in the Bax/Bcl-2 ratio. Overall, *C. avidum* R-CH3 was seen to reduce ROS generated by SA by regulating the secretion of various cytokines and classical AhR signaling, and potentially counteracted the effects of TNF-𝛂 and MAPK signaling to reduce the occurrence of apoptosis.

Next, the expression of genes in the Nrf2 signaling pathway, such as *NFE2L2*, *NQO1*, and *HMOX1,* was analyzed as a possible compensatory mechanism for reversing the elevated levels of intracellular ROS (Figure 4c). Expression of *NFE2L2* is essential for initiating Nrf2-dependent signal transduction. It was significantly lower in the SA-treated group compared to normal HaCaT cells. In contrast, the expression of *NFE2L2* was significantly higher in the *S. hominis* R9 and *C. avidum* R-CH3 treated groups compared to the SA treatment group. Downstream genes *NQO1* and *HMOX1* were also expressed consistent with *NFE2LE*, and the *C. avidum* R-CH3 treatment group showed significantly higher *NQO1* and *HMOX1* expression than the *S. hominis* R9 treatment group.

### 2.6. C. avidum R-CH3 Induces Epidermal Terminal Differentiation via the AhR Signaling Pathway

Based on the oxidation/antioxidant status of the ligand, activation of AhR leads to increased Nrf2 signaling transduction that neutralizes oxidative stress [46]. The AhR signaling axis also stimulates the expression of OVOL-1 like 1 (OVOL1) transcription factor [47]. *FLG* and *LOR* are both regulated by the AHR-OVOL1 pathway, but *IVL* appears to be upregulated by AhR independently of OVOL1 [34]. Based on the observation that *C. avidum* R-CH3 can regulate AhR signaling activity, we first compared the expression of *FLG* and *LOR* genes, which are epidermal differentiation factors, based on *S. hominis* or *C. avidum* abundance in the AD and Pso cohorts. Moreover, we analyzed whether *CLDN1*, one of the essential components of the epidermal tight junction, which forms a dynamic pericellular barrier of the epidermis, is also affected by *S. hominis* and *C. avidum* abundance. Both AD and Pso cohorts of the *S. hominis* high group did not show significant differences in expression of any of the three genes (Appendix A). In contrast, the *C. avidum* high group demonstrated significantly higher *FLG* and *LOR* expression than the *C. avidum* low group in the Pso cohort. We also found that the *C. avidum* high group had significantly increased expression of *CLDN1* in the Pso cohort, similar to *FLG* and *LOR* (Figure 4d).

*C. avidum* R-CH3 was next evaluated to determine whether it modulates AhR directly by partly blocking AhR signaling or modulates CYP1-related gene expression, such as *CYP1A1* for antioxidant activity and OVOL1 axis activity through the AhR signaling pathway. Initially, we silenced *AHR* and examined the correlation between the initiation of Nrf2 signaling and the onset of OVOL1 axis activity according to AhR activity by the SA treatment group (Figure 4e). We observed significant reductions in *AHR* gene expression in the SA and *C. avidum* R-CH3 treated groups and in the SA-alone group compared to si-Control when *AHR* was silenced. Furthermore, the expression of *CYP1A1*, a gene downstream of *AHR*, was significantly suppressed when the *AHR* gene was silenced, regardless of treatment with SA alone or in combination with *C. avidum* R-CH3. After si-*AHR* treatment, the expression of *NFE2L2* and *OVOL1* significantly decreased in all groups.

We conducted additional analyses by silencing *CYP1A1* to determine the effect on the expression of *AhR*, *CYP1A1*, *NFE2L2*, and *OVOL1*. Moreover, to determine whether *C. avidum* R-CH3 could directly regulate the expression of downstream genes of AhR signaling, we silenced *CYP1A1* in this treatment group. We analyzed the expression of *AHR*, *CYP1A1*, *NFE2L2*, and *OVOL1* genes in the same manner (Appendix A). In contrast to previous results, a significant increase in *AHR* expression was observed when si-*CYP1A1* was added to both SA alone and SA with *C. avidum* R-CH3 treatment groups. *NFE2L2* and *OVOL1* expression was significantly increased with an increase in *AHR*, regardless of *CYP1A1* silencing.

Lastly, we assessed the differences in gene expression of epidermal differentiation factors, such as *FLG*, *LOR*, *IVL*, and *CLDN1,* with attenuation of AhR signaling (Figure 4f). We observed that *FLG* and *LOR* expression was reduced by about half in the SA-treated group following *AHR* silencing, while *IVL* expression was reduced more than *FLG* and *LOR*. Furthermore, when *C. avidum* R-CH3 was added to the SA-treated group, the expression of *FLG* and *IVL*, but not *LOR*, increased significantly. In contrast, we found that the expression of *CLDN1*, which codes for a critical protein that forms skin-tight junctions, did not differ significantly from that of si-Control during *AHR* silencing and was significantly decreased following SA treatment. Additionally, *C. avidum* R-CH3 reversed the SA-induced decrease in *CLDN1* expression.

## 3. Discussion

Dysbiosis of the skin microbiota can adversely affect skin homeostasis, and the predominant species SA in AD and Pso correlates with gene expression associated with the diseases [48]. Moreover, it is widely recognized that both AD and Pso are inflammatory skin diseases, and while different pathways are involved in the progression of AD and Pso, most of the dysregulated genes in the two diseases are shared [35]. Hence, we present the findings of this study, which examines the mechanisms underlying the skin barrier function regulated by AhR and the therapeutic potential of SA-derived inflammatory recovery.

First, we demonstrated that the reduction of functional microbiomes due to SA predominance suggests the possibility of distinct common immunological mechanisms involved in the progression of the two diseases. Chronic SA is caused by Th2-biased immunity, resulting in a dysfunctional skin barrier. A compromised skin barrier promotes frequent colonization by SA, compromising other functional barriers, including FLG, LOR, IVL, and CLDN1 [49]. Therefore, we identified potential therapeutic skin microbes, such as *S. hominis* R9 and *C. avidum* R-CH3, that inhibit SA colonization. Moreover, these microbes inhibited the biofilm formation of SA and reduced SA colonization in keratinocytes. According to the findings of this study, the reduction in the expression of skin barrier-forming factors, such as FLG, LOR, IVL, and CLDN1, caused by SA colonization can be reversed via the AhR signaling pathway. Patients with AD and Pso may exhibit impaired AhR signaling in SA-induced conditions, resulting in type 2 immune deviation through Th2 cell activation via type 2 chemokines [50,51].

There is no clear understanding of how SA-induced activation of AhR and downstream signaling accelerates keratinocyte differentiation. As shown in Figure 4, although there were no significant differences in *AHR* expression between the two treatment groups based on the abundance of SA, there were differences in *ARNT* expression in establishing the AhR signaling pathway. This could explain ROS accumulation in keratinocytes due to the significantly increased expression of CYP1 family genes (*CYP1A1*, *CYP1A2*, *CYP1B1*) and *PTGS2* [45]. This accumulation of intracellular ROS may increase AhR-CYP1A1-mediated oxidative stress [52], which may further induce the production of pro-inflammatory cytokines, such as IL-1, IL-4, IL-6, IL-8, and IL-13 [29,53]. Our findings suggest that the *S. hominis* and *C. avidum* low groups may contribute to the development of AD and Pso through similar pathophysiological immune networks. Further, IFN- γ response, TNF- α signaling through NF-κb signaling, and inflammatory response are key mechanisms in the progression of both diseases associated with SA skin infection. Specifically, it is possible that TNF-α and IFN-γ signaling may cause increased expression of the mitochondrial *Bax* gene in keratinocytes, resulting in mitochondrial apoptosis [54,55]. Eventually, through the breakdown of the skin barrier, SA colonization may disrupt the AhR signal transduction system [56,57], leading to a cascade of inflammatory responses and apoptosis induced by excessive intracellular oxidative stress [58]. Therefore, this implies that the change in abundance of functional microbiomes by SA colonization plays an important role in the immune responses in AD and Pso.

It is also interesting to note that, as suggested by our data, AhR/ARNT signaling can regulate the progression of AhR signaling and the expression of AhR/Nrf2 signaling and the OVOL1 transcription factor [32,33]. *C. avidum* R-CH3 stimulates *AHR* expression and inhibits intracellular ROS generation via Nrf2 signaling, critical for promoting epidermal terminal differentiation through *OVOL1* and skin barrier repair [46]. However, *AHR* silencing demonstrated that SA is directly involved in expressing both *AHR* and *CYP1A1*, which are classical AhR signaling components. However, it can be observed that *CYP1A1* silencing could not modify *AHR*, *NFE2L2*, and *OVOL1* expression after SA treatment. These results suggest that the expression of *NFE2L2* is regulated independently of *CYP1A1* and that Nrf2 signaling may be triggered by various immune factors derived from SA in addition to ROS generation following classical AhR signaling. In addition, the findings of this study suggest that *C. avidum* R-CH3 can contribute directly to the activation of AhR-related genes, suggesting that the expression of *CYP1A1*, *NFE2L2*, and *OVOL1* is regulated according to AhR activation signals. Therefore, we can infer that *C. avidum* R-CH3 does not directly regulate *CLDN1* expression via the AhR-OVOL1 axis, but rather *C. avidum* R-CH3 suppresses SA colonization to decrease the cascade of immune responses and generation of ROS. Based on these results, we concluded that *C. avidum* could enhance the skin barrier function by modulating AhR signaling.

Based on the increased severity of SA biofilm colonization in epidermal keratinocytes, increased IL-4 and IL-13 bind to the IL-4α1/IL-13Rα1 receptors and activated downstream Janus family tyrosine kinases (JAKs), leading to STAT3/STAT6 phosphorylation [42]. The progression of this mechanism results in the downregulation of genes that function as skin barriers. For example, AhR-mediated upregulation of FLG, LOR, and IVL is inhibited by activation of the IL-13/IL-4-JAK-STAT6/STAT3 axis [27]. In this study, we hypothesized that STAT6 activation in response to IL-13 increases IL-24 expression in keratinocytes, suppressing FLG expression via STAT3 [42]. The activation of STAT6 by IL-13 induces keratinocytes to increase the expression of IL-24, suppressing FLG by activating STAT3 [59,60]. Furthermore, IL-4/IL-13 signaling could interact with group 2 innate lymphoid cells and Th2 cells to impair AhR-mediated FLG expression and promote skin barrier dysfunction by activating OVOL1 [61,62].

We reported that *C. avidum* R-CH3 and *S. hominis* R9 significantly inhibited SA colonization, thus reducing cytokines, such as TSLP, IL-25, and IL-33 produced in the barrier-disrupted epidermis. The increase in expression of TSLP, which is referred to as the biggest problem after SA colonization [20], promotes a Th2 cell population in skin keratinocytes, which produce IL-4, IL-5, and IL-13. In response to SA colonization, Th2 cytokines are released, including IL-5, IL-25, and IL-33 [20], leading to the proliferation of eosinophils, synthesis of serum IgE, differentiation, and chemotaxis, resulting in the differentiation of DCs into Th2 cells [63]. It is suggested that when the composition ratio of *C. avidum* is lowered, activated DCs may cause an imbalance of Th1, Th2, and Th17 cells. Hence, functional skin microbiome species *C. avidum* R-CH3 and *S. hominis* R9 prevent SA colonization, effectively inhibiting the overlapping bioactivity between TSLP, IL-25, and IL-33. Therefore, inhibiting SA colonization and dominance by using components of the functional microbiome, such as *C. avidum* R-CH3 and *S. hominis* R9, can prevent type 2 immune deviation by reducing the production of TSLP, IL-25, and IL-33, and reversing skin barrier dysfunction [10,64]. Additionally, *S. hominis* R9 and *C. avidum* R-CH3 demonstrated a common mechanism that could be targeted to inhibit eosinophil stimulation and mast cell activation by SA bioproducts and prevent basophil degranulation [10].

Finally, we observed that *C. avidum* R-CH3 and *S. hominis* R9 reduced *IL-17* and *IL-22* expression in response to the overexpression of *IL-4* and *IL-13* following SA colonization. As a result, functional microbiomes can inhibit STAT3 activation by reducing the production of IL-4/IL-13 resulting from SA colonization, thus inhibiting the activation of NF-κB and MAPK mechanisms via inhibiting IL-17A expression [25,65]. This suggests that the reduced abundance of *C. avidum* in the skin may have led to an increase in ROS production, which resulted in a spike in TNF-α signaling and apoptosis through MAPK signaling. By these mechanisms, chronic colonization by SA may result in increased IL-17/IL-22 signaling, resulting in the activation of Th17/Th22 cells due to excessive IL13/IL-4 signaling [35]. This is because IL-17A can activate STAT3 without directly activating the JAK-STAT pathway, and IL-22 can activate JAK1/TYK2 and STAT3 to activate NF-κB and MAPK signaling transduction [65]. Our data also indicated that IL-22, caused by SA colonization, has higher mRNA expression levels than IL-17A. Various reports suggest that IL-22, as opposed to IL-17A, contributes more to the development of chronic lesions and excess type 2 deviations in inflammatory skin diseases derived from SA [10,66,67]. Thus, based on the reduced expression of the IL-17 family and IL-22 induced by the functional microbiome, it can be speculated that reduced SA colonization reduced the activity of Th17 and Th22 cells. After SA colonization, major Th1 markers, such as IFN-γ, are commonly expressed in chronic skin lesions [68]. Through activation of the IL-17A/IL-22 axis and subsequent activation of the IL-23/IL-17A axis, we may be able to explain the mechanism underlying the pathogenesis of chronic AD and Pso. The inhibition of SA colonization using a functional microbiome also significantly reduced the expression of IFN-γ, TNF-α, and IL-1β, inhibiting Th2 cell differentiation and upregulating the expression of skin barrier factors.

In summary, in the present study, we reported that the IL-13/IL-4-STAT6/STAT3 axis and AhR signaling could inhibit SA colonization, and investigated the regulatory mechanism and recovery of skin barrier-related proteins using a functional microbiome. The results presented demonstrate that SA colonization leads to activation of the IL-13/IL-4-JAK-STAT6/STAT3 axis, resulting in barrier dysfunction and associated immune responses. In particular, SA can activate the AhR-ARNT system and enhance the terminal differentiation of epidermal keratinocytes by stimulating AhR-Nrf2-mediated antioxidant activity. Furthermore, this mechanism was explored for its potential to reduce inflammation induced by excessive IL-13/IL-4 signaling. Using functional microbiomes, it was also shown that the regulation of the signaling mechanisms of the AhR/ARNT system was also directly related to immune regulation in SA-derived Th17/22 and T regulatory cell maturation, which may be crucial for treating chronic AD and Pso in the future. Unfortunately, the metabolites that served as AhR ligands in our study were not identified, indicating that studies identifying the additional molecules targeting the AhR system are needed. It would also be beneficial to understand the relationships between AD and other pathological mechanisms of Pso, as well as other metabolites and species from the functional microbiome.

## 4. Materials and Methods

### 4.1. Transcriptomic Data Analysis

Gene Expression Omnibus (GEO) data used in this study were obtained from accession number GSE121212 in the GEO repository [35]. FASTP was used to quality filter the raw data, adapter trim, and adapter trim with the default parameters [69]. In addition, the filtered data were mapped by HISAT2 to the human database (GRCh38) with default settings [70]. After HISAT2 files were generated, they were separated into mapped and unmapped files with SAMtools’ ’view’ and ‘sort’ commands. After that, the unmapped BAM files were converted into unmapped FASTQ files using SAMtools with the ‘bam2fq’ command [71]. To extract taxonomic profiles from the unmapped FASTQ files, we used Kraken2 with the Minikraken database. In the course of this process, we used the commands ‘--use-names’, ‘--gzip-compressed’, ‘--use-map-style’, and ‘--report-zero-counts’ [72]. For contamination removal, packages ‘decontam’ and ‘phyloseq’ in R version 4.1.2 were used [73,74]. The ‘DESeq2’ package was then used to normalize the data and determine differential abundance [75]. Transcriptomic data were processed with featureCounts using default settings to extract gene expression information [76]. As with Kraken2 outputs, decontamination and normalization were carried out using the R studio packages ‘decontam’, ‘phyloseq’, and ‘DESeq2’. GSEA was performed using the GSEA platform version 4.1.0 (Massachusetts Institute of Technology, MA, USA) with the hallmark gene set in MSigDB after normalization. Digital cytometry was conducted using the CIBERSORTx platform (https://cibersortx.stanford.edu, (accessed on 14 May 2022)) [36].

### 4.2. Human Skin Sample Collection

The Korea Institute for Bioethical Policy (KONIBP) Institutional Review Board (IRB; P01-201605-31-003) approved the collection of skin samples and isolated bacteria from human skin, and all protocols complied with relevant ethical guidelines. Further, all participants gave written informed consent prior to registration, and the study followed applicable ethical guidelines. All skin samples were collected from healthy donors without a history of skin disease. Skin samples were collected from participants after they had not washed their faces for more than 12 h. To collect skin bacteria, a sterile cotton swab was moistened with distilled water and rubbed on the face skin vigorously for 1 min or 20 times, then incubated in 10 mL of tryptic soy broth (TSB) solution or Robertson’s cooked meat (RCM) solution.

### 4.3. Microbial Sample Isolation and 16S rRNA PCR Amplification

Skin samples from 20 donors were diluted 10^−1^–10^−3^ fold using phosphate-buffered saline (PBS). Then, 100 µL of each diluted solution was spread onto TSB, RCM, MRS (De Man, Rogosa, and Sharpe), and blood (Columbia agar with 5% sheep blood) agar plates (Bio-Rad, Hercules, CA, USA). The inoculated plates were incubated at 37 °C for up to 72 h, after which colonies were retrieved, and their 16S rRNA genes were amplified using colony PCR. The PCR cycling conditions were as follows: initial denaturation at 95 °C for 15 min, followed by 32 cycles of denaturation at 95 °C for 30 s, annealing at 55 °C for 30 s, extension at 72 °C for 1 min 45 s, and a final extension step at 72 °C for 5 min. The primers for PCR were 27F (5′-AGAGTTTGATCCTGGCTCAG-3′) and 1492R (5′-GGTTACCTTGTTACGACTT-3′). Purification of the amplified DNA was carried out using the Ez-pure PCR purification kit (Ver 2, Enzynomics Co. Ltd., Daejeon, Korea), and the nucleotide sequences of the genes were determined using the ABI 3730xl system (Macrogen Inc., Seoul, Korea). For phylogenetic analysis, the 16S rRNA gene sequences were analyzed using the nucleotide BLAST program available at the NCBI website (https://blast.ncbi.nlm.nih.gov/Blast.cgi?PROGRAM=blastn&PAGE_TYPE=BlastSearch&LINK_LOC=blasthome, (accessed on 1 December 2021)).

### 4.4. Measurement of S. aureus KCTC 1621 Biofilm Formation Inhibition

*Staphylococcus aureus* KCTC 1621 used in this study was distributed from Korean Collection for Type Cultures (KCTC, Jeongeup, Korea). The culture of *S. aureus* was performed in a liquid medium (TSB + 0.2% glucose) for 16 to 24 h. Once TSB containing 0.2% glucose was placed in a 6-well plate, each bacterial supernatant group was added to each well at a volume of about 10%. In each well, 2 × 10^6^ CFU were inoculated so that a final concentration of 2 × 10^6^ CFU/well was achieved. Cells were then cultured at 37 °C in an incubator for 24 h. Following incubation, the medium was removed, and each well was washed twice with 1–2 mL of sterile PBS. Biofilms of *S. aureus* were scraped off with a scraper, and the absorbance was measured at 600 nm after being well suspended. A BioPhotometer D30 (Eppendorf Inc., Hamburg, Germany) was used to measure the absorbance. To determine the ability to inhibit biofilm formation, we used untreated controls as negative controls, and wells inoculated with baicalein (25 µg/mL) as positive controls.

### 4.5. Growth Inhibition (Overlay Clear Zone) Test for S. aureus KCTC 1621

The overlay clear zone assay was performed to analyze the effects of various bacteria isolated from normal skin samples on growth inhibition of *S. aureus*. *C. avidum* R-CH3 medium was plated onto a thin RCM agar plate and cultured for 72 h at 37 °C. Once *C. avidum* R-CH3 was confluent, *S. aureus* KCTC 1621, adjusted to 10^4^ CFU/mL, was inoculated into 10 mL of RCM agar that had not yet solidified at 45 °C and was sufficiently suspended before the medium was hardened. A hardened agar plate was anaerobically cultured at 37 °C for 40 h for *S. aureus* KCTC 1621 and 72 h for *C. avidum* R-CH3 to determine the size of the clear zone surrounding *C. avidum* R-CH3. PBS served as a negative control, while triclosan served as a positive control.

### 4.6. Cell Culture and Treatment of Several Bacterial Supernatant Solutions

The human skin keratinocyte cell line, HaCaT, was purchased from PromoCell (Heidelberg, Germany). HaCaT cells were maintained at 37 °C in an incubator with a humidified atmosphere of 5% CO_2_. HaCaT cells were cultured in DMEM supplemented with 10% heat-inactivated fetal bovine serum and an antibiotic-antimycotic solution (100 units/mL penicillin, 100 µg/mL streptomycin, and 0.25 µg/mL amphotericin B) (Gibco, Waltham, MA, USA). For *S. aureus* treatment, 2 × 10^5^ HaCaT cells were seeded in 6-well plates, incubated in an atmosphere of 5% CO_2_ at 37 °C, and cultured until they achieved 80% confluence. After 12 h, the cells were washed once with PBS. An *S. aureus* dilution (adjusted to 10^7^ CFU/mL) was inoculated by replacing the culture medium with a fresh medium without antibiotics. Then, the 10% conditioned bacterial supernatant was added to the cells with a supplement-free medium at the same time as the *S. aureus* treatment.

### 4.7. Total RNA Isolation and qRT-PCR Analysis

Total RNA was isolated from HaCaT cells using TRIzol reagent (TaKaRa, Shiga, Japan) according to the manufacturer’s instructions. In addition, cDNA was synthesized from 1 µg total RNA using Transcription Premix (Elpis-Biotech, Daejeon, Korea) under the following reaction conditions: 45 °C for 45 min and 95 °C for 5 min. Gene expression was quantified using qRT-PCR, and data were analyzed using StepOne Plus^TM^ software (Applied Biosystems, Foster City, CA, USA). qRT-PCR amplification reactions were performed using SYBR Green PCR Master Mix with premixed ROX (Applied Biosystems). The primer pairs (Bioneer, Daejeon, Korea) were used in the ABI 7300 Cycler (Thermo Fisher, Waltham, MA, USA) internal reaction according to the manufacturer’s protocol. The reaction conditions were as follows: 40 cycles for 2 min at 50 °C, 10 min at 95 °C, 10 s at 95 °C, and 1 min at 60 °C. 18S rRNA was used as an internal control.

### 4.8. Measurement of Pro-Inflammatory Cytokines

The pro-inflammatory cytokines IFN-γ, TNF-α, and IL-1β were measured using an ELISA assay. HaCaT cells (6 × 10^5^) seeded in 6-well plates were pretreated with *S. aureus* (1 × 10^7^ CFU/mL) for 12 h and then treated with bacterial supernatants for 12 h. Then, aliquots of samples (100 µL/well) were harvested from the experimental medium, and the production of cytokines was measured using a Human ILs Quantikine ELISA kit (R&D systems, Minneapolis, MN, USA) according to the manufacturer’s instructions.

### 4.9. AHR and CYP1A1 Gene Knockdown

siRNA against human *AHR* and *CYP1A1* mRNA were synthesized commercially at Bioneer: si-*AHR* (forward, 5′-CACUCAGACUACCACACAU-3′, reverse, 5′-AUGUGUGGUAGUCUGAGUG-3′); si-*CYP1A1* (forward, 5′-GCUAGGGUUAGGAGGUCCU-3′, reverse, 5′-AGGACCUCCUAACCCUAGC-3′). Lipofectamine^TM^ RNAiMAX (Invitrogen, Carlsbad, CA, USA) was used to transfect siRNA oligos according to the manufacturer’s instructions. A total of 3 × 10^5^ HaCaT cells were seeded into 6-well plates containing 2.5 mL of antibiotic-free growth medium 24 h prior to transfection to reach a maximum of 60% confluence at the time of transfection. Each siRNA was incubated with HaCaT cells for 24 h, and the efficiency of gene silencing was evaluated using qRT-PCR.

### 4.10. Statistical Analysis

All data were tested for normality, and datasets were analyzed using one-way or two-way analysis of variance. Post-hoc analysis was then carried out using the Bonferroni test or Tukey’s test for comparison between pairs. Data are presented as the mean ± standard error of the mean. All statistical analyses were performed using GraphPad Prism 9 (GraphPad Software Inc., La Jolla, CA, USA) and R-4.2.0 for Windows. For comparisons of DEGs, the fold change value was set at greater than 2, and the *p* value at less than 0.05. To compare the two groups, we used the Mann-Whitney–Wilcoxon test. Statistical significance was set at *p* < 0.05.

## Figures and Tables

**Figure 1 ijms-23-09551-f001:**
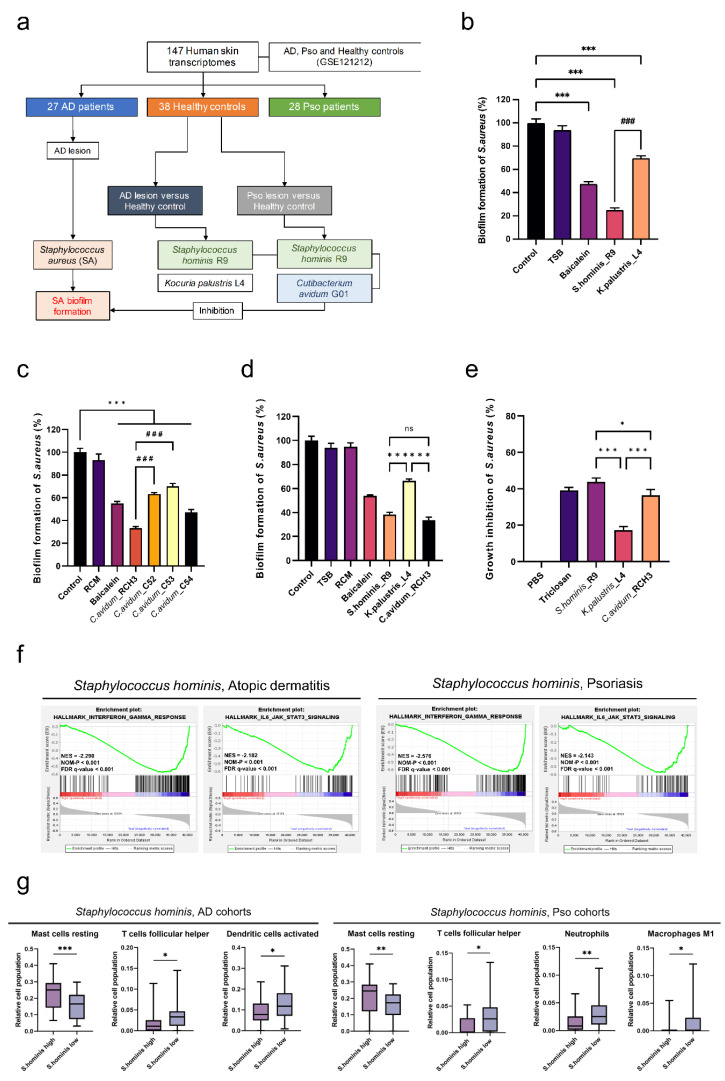
Potential therapeutic microbiomes are capable of inhibiting *Staphylococcus aureus* (SA)-derived biofilms that may initiate atopic dermatitis (AD) and psoriasis (Pso) pathogenesis. (**a**) Schematic of GSE121212 transcriptome analysis and skin microbiome isolation. (**b**) Evaluation of *S. hominis* R9 and *K. palustris* L4 to inhibit SA biofilm formation. *** *p* < 0.001 compared to control group. (**c**) Evaluation of SA biofilm formation inhibition ability by *C. avidum* strains. *** *p* < 0.001 compared to control group. ### *p* < 0.001 compared to *C. avidum* R-CH3 group. (**d**) Comparing inhibition of SA biofilm formation by *S. hominis* (r9), *C*. *avidum* (R-CH3), and *K*. *palustris* (l4). *** *p* < 0.001 compared to *K. palustris* L4 group; ns, non-significant. (**e**) Evaluation of SA growth inhibition ability of *S. hominis* (r9), *C. avidum* (R-CH3), and *K. palustris* (l4). *** *p* < 0.001 compared to *K. palustris* L4 group; # *p* < 0.05 between *S. hominis* R9 and *C. avidum* R-CH3 groups. (**f**) Gene set enrichment analysis (GSEA) stratified by *S. hominis* abundance in AD and Pso cohorts (nominal [NOM] *p* < 0.001; interferon gamma [IFN-γ] response, [NOM] *p* < 0.001; IL6-JAK-STAT3 signaling in AD cohort, [NOM] *p* < 0.001; IFN-γ response, [NOM] *p* < 0.001; IL6-JAK-STAT3 signaling in Pso cohort). (**g**) Boxplots comparing macrophages (M1), resting mast cells, follicular T helper cells, dendritic cells, and neutrophils stratified by an abundance of *S. hominis* in AD and Pso. * *p* < 0.05, ** *p* < 0.01, *** *p* < 0.001 comparing *S. hominis* high versus low group. Statistical significance was calculated using Bonferroni tests. CIBERSORT algorithm was used for the Cancer Genome Atlas (TCGA) data of AD and Pso cohorts.

**Figure 2 ijms-23-09551-f002:**
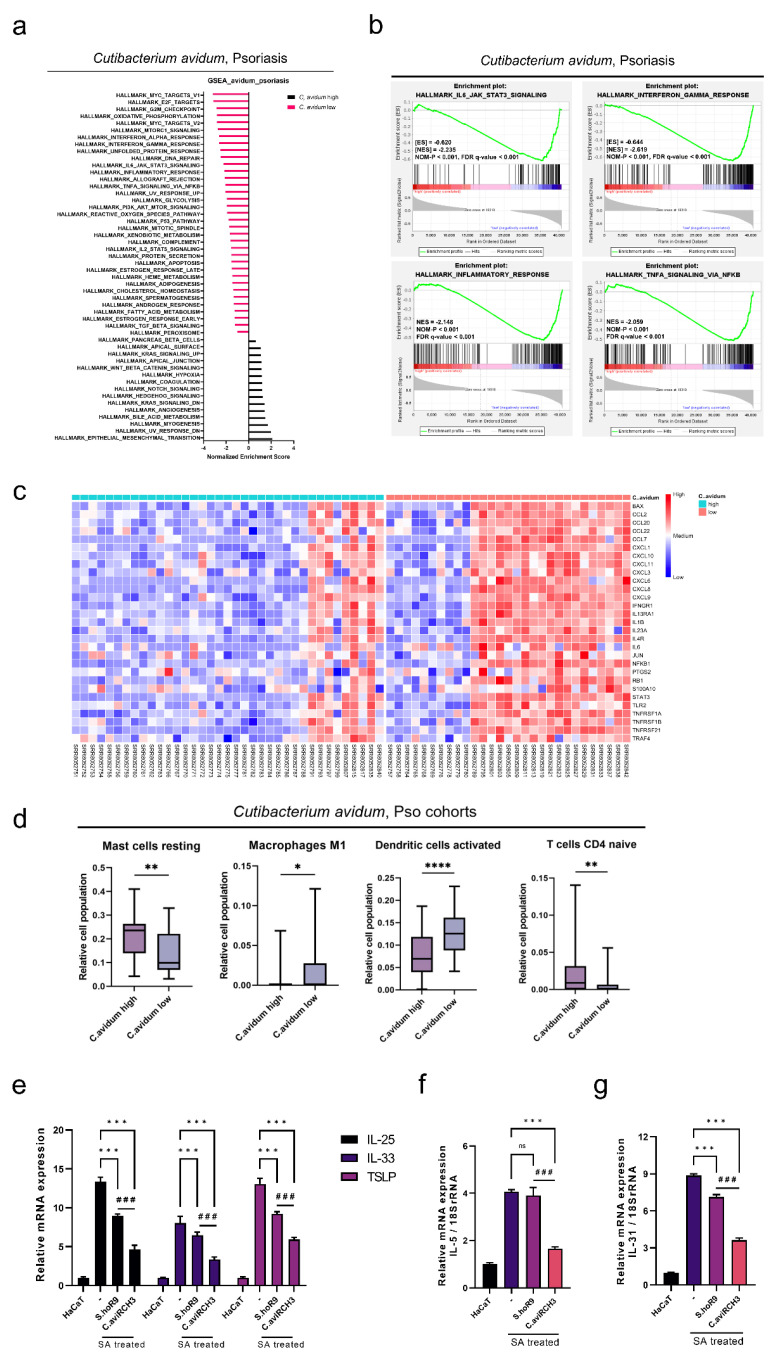
Characterization of AD and Pso-induced immune modifications concerning functional microbiome abundance and immune suppression mediated by SA-derived mechanisms. (**a**) Normalized enrichment scores (NES) of gene set enrichment analysis (GSEA)-based gene sets stratified by *C. avidum* abundance in a Pso cohort. (**b**) The top four gene sets of the *C. avidum* high group in GSEA. ([NOM] *p* < 0.001; IL6-JAK-STAT3 signaling, [NOM] *p* < 0.001; IFN-γ response, [NOM] *p* < 0.001; inflammatory response, [NOM] *p* < 0.001; TNF-α signaling via NF-κB). (**c**) Heatmap representing significant immune-related genes in the analysis of differentially expressed genes, comparing *C. avidum* high versus low groups. (**d**) Boxplots comparing macrophages (M1), resting mast cells, activated dendritic cells, and naïve CD4 T cells accordingly stratified by the abundance of *C. avidum* in Pso. * *p* < 0.05, ** *p* < 0.01, **** *p* < 0.001 comparing *C. avidum* high versus low groups. (**e**) Measurement of mRNA levels of *IL-25*, *IL-33*, and *TSLP* relative to 18S rRNA. *** *p* < 0.001 compared to SA treatment group. ### *p* < 0.001 between *S. hominis* R9 and *C. avidum* R-CH3 groups. (**f**) Measurement of mRNA level of *IL-5* relative to 18S rRNA. *** *p* < 0.001; ns: not-significant compared to SA only treatment group. ### *p* < 0.001 between *S. hominis* R9 and *C. avidum* R-CH3 groups. (**g**) Measurement of mRNA level of *IL-31* relative to 18S rRNA. *** *p* < 0.001 compared to control group. ### *p* < 0.001 between *S. hominis* R9 and *C. avidum* R-CH3. Statistical significance was calculated using Bonferroni tests. CIBERSORT algorithm was used for the TCGA data of the Pso cohort.

**Figure 3 ijms-23-09551-f003:**
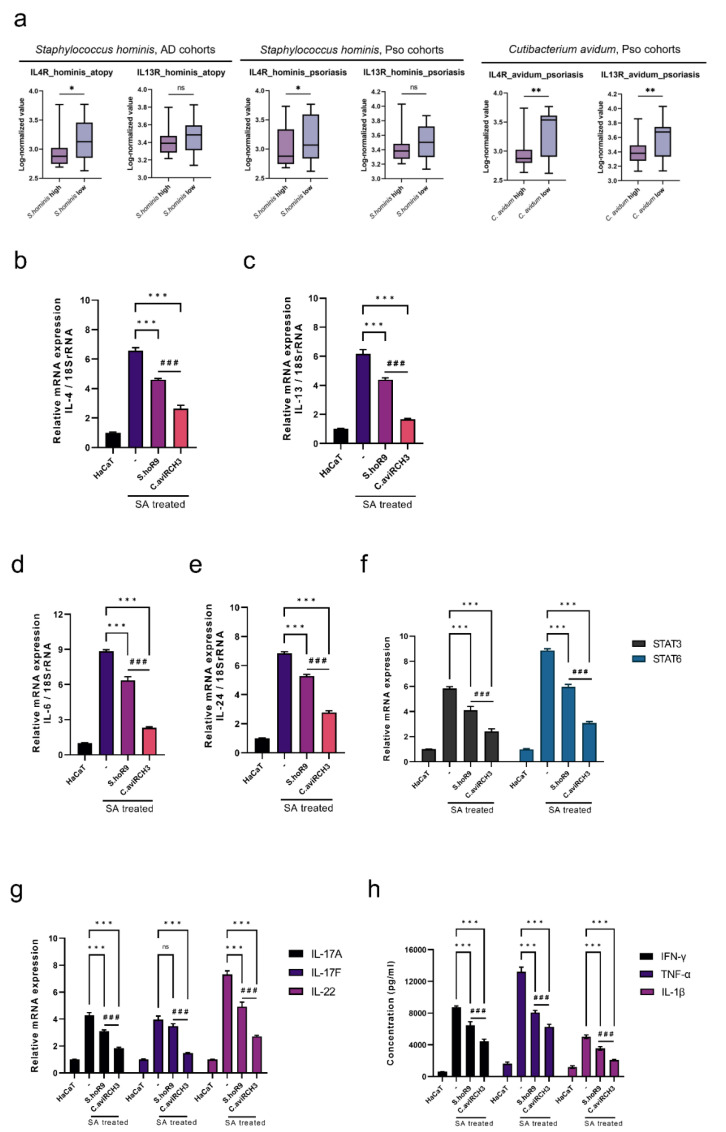
The functional microbiome regulates Th2-biased cytokine immune responses by inhibiting SA-derived IL-4/IL-13 axis activation. (**a**) Boxplots of gene expression values for *IL-4R* and *IL-13RA1* stratified by an abundance of *S. hominis* and *C. avidum* in AD and Pso cohorts. (**b**) Measurement of mRNA level of *IL-4* relative to 18S rRNA. (**c**) Measurement of mRNA level of *IL-13* relative to 18S rRNA. (**d**) Measurement of mRNA level of *IL-6* relative to 18S rRNA. (**e**) Measurement of the mRNA level of *IL-24* relative to 18S rRNA. (**f**) Measurement of mRNA levels of *STAT3* and *STAT6* relative to 18S rRNA. (**g**) Measurement of mRNA levels of *IL-17A*, *IL-17F,* and *IL-22* relative to 18S rRNA. (**h**) Measurement of the concentrations of IFN-γ, TNF-α, and IL-1β cytokines using ELISA after treatment with *S. hominis* R9 and *C. avidum* R-CH3 derived microbial supernatants in the SA treatment group. * *p* < 0.05, ** *p* < 0.01, *** *p* < 0.001 compared to SA only treatment group. ### *p* < 0.001 between *S. hominis* R9 and *C. avidum* R-CH3. ns: not-significant compared to; ns: not-significant compared to SA only treatment group. Statistical significance was calculated using Bonferroni tests.

**Figure 4 ijms-23-09551-f004:**
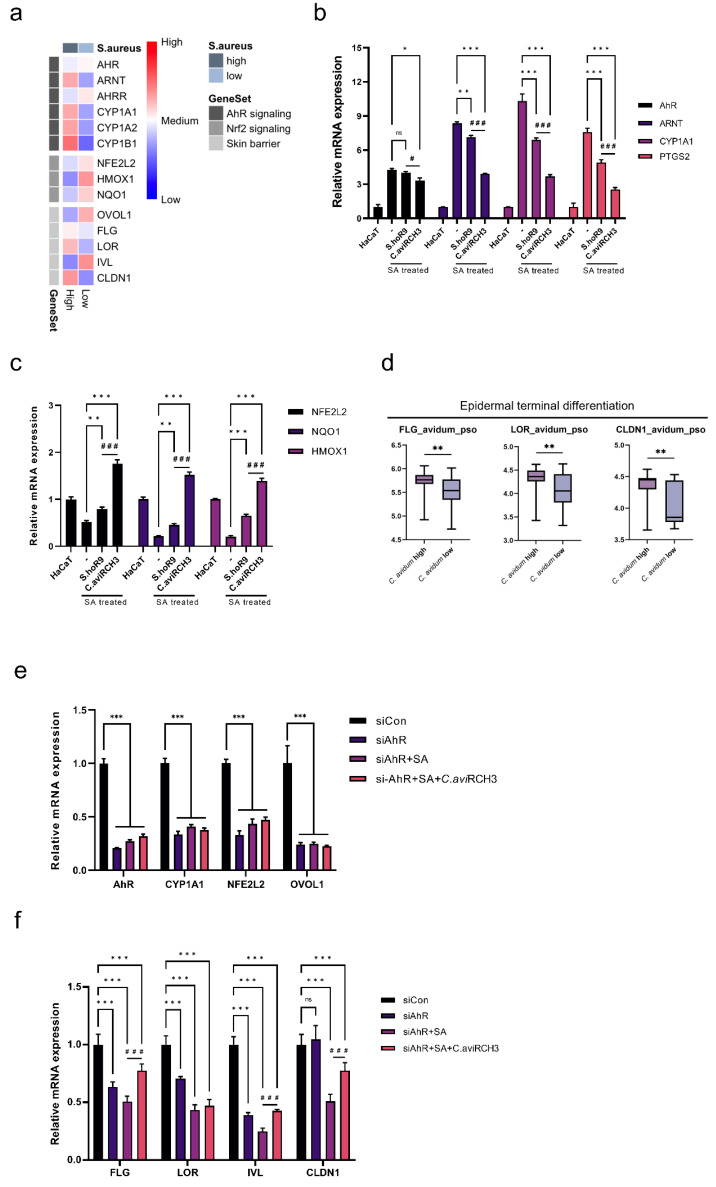
Functional microbiomes modulate ROS-induced keratinocyte apoptosis via dual AhR/Nrf2 signaling and induce epidermal terminal differentiation. (**a**) Heatmap showing AhR signaling, Nrf2 signaling, and skin barrier-related genes stratified by SA abundance. (**b**) Measurement of mRNA levels of *AhR*, *ARNT*, *CYP1A1,* and *PTGS2* relative to 18S rRNA. * *p* < 0.05, ** *p* < 0.01, *** *p* < 0.001 compared to control group (HaCaT). # *p* < 0.05, ### *p* < 0.001 between *S. hominis* R9 and *C. avidum* R-CH3. (**c**) Measurement of mRNA levels of *NFE2L2*, *NQO1,* and *HMOX1* relative to 18S rRNA. ** *p* < 0.01, *** *p* < 0.001 compared to control group (HaCaT). ### *p* < 0.001 between *S. hominis* R9 and *C. avidum* R-CH3. (**d**) Boxplots of gene expression values for *FLG*, *LOR,* and *CLDN1* stratified by an abundance of *C. avidum* in the Pso cohort. ** *p* < 0.01 between *S. hominis* R9 and *C. avidum* R-CH3. (**e**) Measurement of mRNA levels of *AHR*, *CYP1A1*, *NFE2L2,* and *OVOL1* relative to 18S rRNA in SA and SA and *C. avidum* treated groups with or without *AHR* gene silencing in HaCaT cells. *** *p* < 0.001 compared to each si-control group. (**f**) Measurement of mRNA levels of *FLG*, *LOR*, *IVL,* and *CLDN1* relative to 18S rRNA in SA and SA and *C. avidum* treated groups with or without *AHR* gene silencing in HaCaT cells. *** *p* < 0.001, ns: not-significant compared to each si-control group. ### *p* < 0.001 between SA treatment group and *C. avidum* R-CH3 treatment group after SA treatment in *AHR*-silenced HaCaT cells. Statistical significance was calculated using Bonferroni tests.

**Table 1 ijms-23-09551-t001:** List of the skin microbiomes used in this study.

Species	Strains	Abbreviated Name of Bacteria	Closest Match (16S rRNA Gene Similarity)	Hemolysis
** *Staphylococcus hominis* **	WF7R9	*S. hominis*_R9	*Staphylococcus hominis* (99%)	Γ
** *Kocuria palustris* **	WF3L4	*K. palustris*_L4	*Kocuria palustris* (100%)	Γ
** *Cutibacterium avidum* **	R-CH3	*C. avidum*_R-CH3	*Cutibacterium avidum* (99%)	γ
	CSM5-2	*C. avidum*_C52	*Cutibacterium avidum* (100%)	γ
	CSM5-3	*C. avidum*_C53	*Cutibacterium avidum* (99%)	γ
	CSM5-4	*C. avidum*_C54	*Cutibacterium avidum* (99%)	γ

## Data Availability

This study used data processed from Gene Expression Omnibus, accession number GSE121212, to analyze gene expression. This data is freely available at https://www.ncbi.nlm.nih.gov/geo/query/acc.cgi?acc=GSE121212, (accessed on 20 August 2022).

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
