# Peer review of "Potential Therapeutic Skin Microbiomes Suppressing Staphylococcus aureus-Derived Immune Responses and Upregulating Skin Barrier Function-Related Genes via the AhR Signaling Pathway"

_ijms, 2022, doi:10.3390/ijms23179551_

Round 1

Reviewer 1 Report

The ability of members of the skin microbiota to modulate growth of S. aureus is key to homeostasis in healthy skin. Dysbiosys occurs in atopic dermatitis and psoriasis leading to overgrowth of pathogens (especially S.aureus in AD). The paper examined the ability of members of the skin microbiota to reverse the effects on expression of cytokines and  genes involved in normal barrier function via the AhR signalling pathway.

The paper is expertly composed and is well written in correct succinct scientific English.

1.It is incorrect to use the term “ functional microbiomes” when dealing with individual microbes isolated from the skin microbiome. Furthermore the skin barrier was not measured directly in this study, it is inferred.  Careful wording of the title and the abstract are required

2.The rationale for studying biofilm in the context of skin disease should be stated. Is there any evidence that S. aureus forms biofilm when growing on AD skin ?

3.In the biofilm inhibition experiments described in Figure 1 it is not clear how many S.aureus strains were tested. It seems that a number of skin isolates from diseased and healthy controls donors were used. More information is required

4.The rationale for using baicalein as a control should be given. This inhibits biofilm formation but was not tested in the growth inhibition experiments  

5.Can it be concluded that the organism (Kokuria palustris) with the weakest effect on growth of S.aureus  has the weakest effect on biofilm? Therefore density of  planktonic growth in the supernatant of the biofilm growth chambers  will likely correlate with biofilm density unless there is a specific inhibitor of biofilm formation that does not affect growth per se

Author Response

  • File name: Response to Reviewer's comments_Reviewer 1

Reviewer 2 Report

Lee et al. explored the functional microbiome by focusing on atopic dermatitis (AD)/psoriasis (Pso) pathologies, characterized by impaired epidermal differentiation with opposing immune profiles, namely, Th2 (anti-parasitic responses) vs. Th17 (anti microbial responses).

This study consists of two parts; 1. the bioinformatic analysis that extracted Staphylococcus hominis_R9 and Cutibacterium avidum_H3 as the components of beneficial cutaneous microbiota, including in-silico immune cell subset analyses, and 2. in vitro functional study that examined the immuno-modulatory/tissue-repairing effects when S. aureus (SA)’s toxicity is present, using immortalized HaCaT keratinocytes (KCs).

Overall, this manuscript is well-written and could add evidence that may warrant further investigation on potential therapeutic interventions (microbiome-based therapeutics).

Major issues

1. L348–“transduction of AhR signaling activity by SA seems to be mediated by ARNT and may contribute to the increased expression of CYP1A1 and PTGS2 as well as the generation of ROS.”

The major drawback of this study is that the authors examined the immunological effects of the beneficial species by using only HaCaT KCs; neither primary human KC nor leukocytes and immune cell subset analysis ended up in-silico. 

Both AhR and ARNT have broad arrays or biological effects in vivo. The previous reports, such as Geng et al. J Cell Sci. 2006;119: 4901-12. or Robertson et al. J Cell Sci. 2012;125: 3320-32., suggest the AhR/ARNT axis affects the development of tissue responses that require cross-talking between KCs and leukocytes. 

The authors should incorporate it into limitation statements because more careful interpretation of the data is required. Additionally, if possible, they could present newly analyzed data from the GEO data set, such as tissue regulatory T cells that presumably maintain tolerance against skin microbiota through the amphiregulin-EGFR signaling (such as Zaiss et al. Immunity. 2013;38: 275-84. d), in agreement with the colonization data.  

2. Regarding epidermal differentiation (FLG/LOR), lentiviral ARNT suppression increases these markers (Robertson et al. J Cell Sci. 2012;125: 3320-32.), somewhat conflicting with one of the authors’ conclusions, as in L391– “we concluded that C. avidum can enhance the skin barrier function by modulating AhR signaling” The authors should at least mention the previous study and could reconcile the conflict.

Minor issues

1. L15: “A dysfunctional skin barrier develops in AD and Pso as a result of SA colonization, resulting in tissue fibrosis” This looks confusing. Pso is caused by aberrantly activated innate immunity and rarely develops fibrosis. Revise the passage for clarity.

2. L20: “nuclear factor erythroid-2-related factor 2 (Nrf2) receptors”: NRF2 downstream targets might be better because NRF2 by itself is not a receptor.

3. L208: resting mast cells, rather than just mast cells.

4. L346:PTGS2-derived from SA colonization? L464: Chronic SA colonization? L538:IL-22-derived from SA colonization? L539:IL-17A-derived from SA colonization?

5. L377: What an antioxidant enzyme refers to is unclear; please revise.

7. L448–: the discussion section felt too long and rambling. Make it more concise, if possible.

8. L523–24: It was hard to comprehend what the authors intended to claim.

Author Response

  • File name: Response to Reviewer's comments_Reviewer 2

Reviewer 3 Report

The manuscript entitled “Functional microbiomes regulate skin barrier function via the AhR signaling pathway in response to Staphylococcus aureus-derived immune responses” is well written and well organized by the Authors.

The 'Introduction' section is well balanced and accompanied by a rich and up-to-date bibliography.

The results are manifold and described clearly and in detail by the authors who also processed the same data with an excellent and relevant statistical analysis. This is further clarified by the graphs and figures with which they have enriched the manuscript.

I do not know whether this is possible due to editorial constraints, but it would be convenient to insert most if not all of what is contained in the supplementary material directly into the body of the main manuscript.

The discussion is very interesting and thorough and actually weighs down the manuscript a bit. I would advise the authors to make it more fluent because at times it is difficult to follow, reducing some passages to more concise sentences that are immediately directed to the crux.

Ultimately, I think the manuscript is of excellent quality and, subject to these minor changes, should be published as soon as possible.

Author Response

  • File name: Response to Reviewer's comments_Reviewer 3
